# Genetic Diversity of Tomato Black Ring Virus Satellite RNAs and Their Impact on Virus Replication

**DOI:** 10.3390/ijms23169393

**Published:** 2022-08-20

**Authors:** Julia Minicka, Agnieszka Taberska, Aleksandra Zarzyńska-Nowak, Katarzyna Kubska, Daria Budzyńska, Santiago F. Elena, Beata Hasiów-Jaroszewska

**Affiliations:** 1Department of Virology and Bacteriology, Institute of Plant Protection—National Research Institute, Wł. Węgorka 20, 60-318 Poznan, Poland; 2Instituto de Biología Integrativa de Sistemas (I2SysBio), CSIC—Universitat de València, 46980 València, Spain; 3The Santa Fe Institute, Santa Fe, NM 87501, USA

**Keywords:** tomato black ring virus, satRNAs, genetic variability, RT-qPCR, viral load

## Abstract

Viral satellite RNAs (satRNAs) are small subviral particles that are associated with the genomic RNA of a helper virus (HV). Their replication, encapsidation, and movement depend on the HV. In this paper, we performed a global analysis of the satRNAs associated with different isolates of tomato black ring virus (TBRV). We checked the presence of satRNAs in 42 samples infected with TBRV, performed recombination and genetic diversity analyses, and examined the selective pressure affecting the satRNAs population. We identified 18 satRNAs in total that differed in length and the presence of point mutations. Moreover, we observed a strong effect of selection operating upon the satRNA population. We also constructed infectious cDNA clones of satRNA and examined the viral load of different TBRV isolates in the presence and absence of satRNAs, as well as the accumulation of satRNA molecules on infected plants. Our data provide evidence that the presence of satRNAs significantly affects viral load; however, the magnitude of this effect differs among viral isolates and plant hosts. We also showed a positive correlation between the number of viral genomic RNAs (gRNAs) and satRNAs for two analysed TBRV isolates.

## 1. Introduction

The genome of plant viruses can be often accompanied by a variety of subviral RNAs. These particles are relatively short and non-infectious, and their replication, encapsidation, and spread depend on the helper virus (HV) [1,2]. Two groups of subviral RNAs can be distinguished based on their origin: satellite RNAs (satRNAs) and defective RNAs (D RNAs). SatRNAs share little sequence similarity with the viral genomic RNAs, whereas D RNAs are derived from the genome of the HV, and are mainly formed via RNA recombination/genome rearrangements [1,2]. D RNAs can be also referred to as “interfering” if they interfere with the replication of their HV and affect symptom development. It has been shown that the presence of subviral RNAs might have a great impact on viral replication, accumulation, and symptoms observed in infected plants [3,4,5]. These features make subviral RNAs a good biological system for the study of the molecular function of viruses via replication and translation [6].

Additional RNA molecules have been found in many representatives of the *Nepovirus* genus, being the largest group of plant picorna-like viruses (https://talk.ictvonline.org/taxonomy/, accessed on 4 July 2022). Nepoviruses have been separated into three subgroups (A, B, and C) based on the RNA2 size, phylogenetic relationships of the coat protein gene, and cleavage site specificity of the protease [7]. Tomato black ring virus (TBRV) is a representative of subgroup B that has a wide natural and experimental host range, including herbaceous and woody plants [8,9,10,11]. The TBRV genome consists of two single-stranded RNAs, each of them carrying a small covalently attached VPg protein at their 5′ end and a poly(A) tail at their 3′ end. Each genomic RNA encodes a polyprotein, which is proteolytically cleaved into mature functional proteins by the RNA1-encoded cysteine protease. RNA1 is responsible for viral replication and polyproteins’ processing and RNA2 for encapsidation and movement in plants [12]. The TBRV genome can be accompanied by both D RNAs and satRNAs [4,13,14,15]. It has been shown that D RNAs interfere with TBRV replication, modulating virus accumulation and symptoms observed in infected plants [4]. Two types of satRNAs of nepoviruses were identified: the type B satRNAs, which are 1100–1500 nt, and the type D satRNAs, which are less than 500 nt [7]. TBRV satRNAs consist of 1347 nucleotides, and, like type B satRNAs, are flanked by a VPg at the 5′ end, a poly(A) tail at the 3′ end, and encode a nonstructural protein [16]. The satRNA-encoded protein of 48K has been shown to be involved in the replication of the satRNAs [13]. Although satRNAs and HV do not possess obvious sequence similarity, the association between them seems to be highly specific, as HV supports replication of the particular type of satRNA [17]. The origin of satRNAs is generally unknown, but they may originate by the recombination of viral/and or host nucleic acids [18]. A consensus sequence (UG/UGAAAAU/AU/AU/A) in the 5′ untranslated region (UTR) was found in the satRNA and the genomic RNAs of grapevine fanleaf virus (GFLV) and several other nepoviruses [19]. Recombination events are likely to occur between the 5′ UTR of an ancestral subgroup A nepovirus RNA and an unidentified RNA [18].

Knowledge regarding the role of satRNAs in pathogenesis is still rather limited. Generally, the presence of satRNAs might affect symptom development by attenuating or exacerbating them, however, it can also have no significant effect [1,20]. Most large nepovirus satellites do not influence viral symptoms, and no correlation has been observed so far between symptoms induced by e.g., GFLV or arabis mosaic virus (ArMV) and the presence/absence of satRNAs [19]. In this study, we address the genetic diversity of satRNAs associated with TBRV isolates that originated from different hosts and their effect on HV replication and disease symptoms. We aimed to analyse the satRNAs’ population, its structure, and the phylogenetic relationships between particular satRNAs. We determined the impact of satRNAs on symptom development and TBRV accumulation in three different hosts: quinoa (*Chenopodium quinoa* Willd), tobacco (*Nicotiana tabacum* L. var Xhanti), and spinach (*Spinacia oleracea* L.). Moreover, we analysed the accumulation of satRNAs in all combinations and correlated the accumulation of genomic RNA (gRNA) and satRNAs during the infection.

## 2. Results

### 2.1. Analysis of Genetic Diversity of satRNAs

In 18 among 42 analysed samples, the presence of satRNAs was confirmed (Table 1). Plasmid DNAs containing the full-length sequences of each satRNA were sequenced. For each satRNA, at least three independent clones were sequenced. The obtained satRNAs sequence lengths ranged from 1371 to 1374 nt and included a single open reading frame of 1257 nt, a 5′ UTR of 40 nt, and a 3′ UTR of 72 to 75 nt. Differences in length of satRNAs occurred in the following isolates: sTBRV-Byd4 (1372 nt), sTBRV-Byd8 (1372 nt), sTBRV-E (1372 nt), and sTBRV-CH (1371 nt), and resulted from the deletion of U and A nucleotides at positions 1336 and 1337 (sTBRV-Byd4, sTBRV-Byd8), of C, U, and A nucleotides at positions 1335, 1336, and 1337 (sTBRV-CH), and of C and U nucleotides at positions 1335 and 1336 (sTBRV-E) in the 3′ UTR region. The analysis revealed that the satRNAs are diverse and their nucleotide identity ranged from 91.1% to 100% and from 89.4% to 100% for the amino acid sequences, respectively. The greatest differences were observed between sTBRV-C and sTBRV-O1. For these satRNAs, 111 nucleotide substitutions were found. The obtained sequences of satRNAs were deposited in the GenBank under the accession numbers indicated in Table 1. For further analyses, the coding region of satRNAs was aligned by codons using MUSCLE, as implemented in MEGA X [21].

### 2.2. Recombination and Selective Preasure Analysis of satRNAs

Analyses performed with RDP4 v. 4.100 program [23] and GARD algorithm [24] revealed that there were not recombination events in the data set.

The strength of selection acting on particular codons along the coding region of the satRNAs population was assessed by FEL, FUBAR, SLAC, and MEME algorithms, as implemented in the HyPhy package (https://www.datamonkey.org/, accessed on 20 June 2021). Codons under positive and negative selective pressure are marked in Table 2. Performed analyses revealed the presence of 41, 14, and 44 codons indicated by FEL, SLAC, and FUBAR, respectively, to be under purifying selection (ω < 1); 14 of them were pointed out by all of the used algorithms. Although purifying selection (ω < 1) seems to be the prevailing force shaping the TBRV satRNAs population, overall, 17 codons were detected to be under positive selective pressure (ω > 1). Two of them were identified by three out of the four methods (codons 74 and 224, indicated by FUBAR, FEL, and MEME), three were identified both by FEL and FUBAR (at positions 125, 162, and 244), and four were identified simultaneously by MEME and FUBAR (at positions 19, 227, 245, and 351) (Table 2).

### 2.3. Phylogenetic Analysis of satRNAs

Phylogenetic reconstruction based on nucleotide sequences of TBRV satRNAs revealed the presence of one major clade, which grouped 25 out of 26 analysed satRNAs. Despite the fact that, within the satRNAs population, those associated with TBRV isolates from black locust were the most abundant; they did not comprise a monophyletic group. Instead, they clustered together with satRNAs from zucchini, cucumber, marigold, or lettuce (sTBRV-K8, sTBRV-O1, sTBRV-Ag, and sTBRV-S1, respectively). sTBRV-C, originated from *Apium* sp. in the UK, was the most divergent isolate (Figure 1a). Different TBRV satRNAs grouping present an ML tree constructed based on amino acid sequences of the satRNAs coding region, where two distinct clades were distinguished (Figure 1b). A first cluster was constituted by satRNAs isolated from back locust (*R. pseudoacacia* L.), horseradish (*A. rusticana* Gaertn, Mey & Scherb), cucumber (*C. sativus* L.), zucchini (*C. pepo* L. convar. giromontiina), vervain (*V. officinalis* L.), lettuce (*L. sativa* L.), and French marigold (*T. patula* L.). In a second clade, sTBRV-C clustered with sTBRV-WM1, sTBRV-Sier1, sTBRV-WM2, and sTBRV-WM3, all of which were isolated from black locust, indicating a close relationship between these particles. For both trees, no clear correlation was found between the genetic variation of satRNA, the host species, and the site of origin.

### 2.4. Analysis of the Absolute Quantification of TBRV Genomic RNA in Presence and Absence of satRNAs

All isolates (TBRV-K8, TBRV-M1, and TBRV-P1) caused similar disease symptoms in infected plants, with slightly stronger symptoms caused by TBRV-K8. The first symptoms in infected plants were visible after 10 days post-inoculation (dpi) (Figure 2). In quinoa, symptoms included leaf chlorosis, malformation, and reduction of growth (Figure 2a,d). In tobacco, chlorotic ringspots and mosaic were visible (Figure 2b,e); and in spinach, chlorosis, and reduction of growth were noticed (Figure 2c,f). The presence of the satRNAs had an effect on the magnitude of the symptoms in the case of quinoa and spinach, where the symptoms caused by viral infection were stronger. In the case of tobacco, the symptoms were similar, regardless of the presence or absence of satRNA particles.

The fit of the RT-qPCR data to the repeated measures that ANOVA described in Section 4.3.3 (Equation (1)) showed that TBRV isolate ($\eta_{P}^{2}$ = 0.873, *p* < 0.0001) and host species ($\eta_{P}^{2}$ = 0.982, *p* < 0.0001) both have highly significant and large effects on viral load. The highest log-virus titre was found for TBRV-P1 (estimated marginal mean ±1 SD: 8.732 ± 0.003), followed by TBRV-K8 (8.691 ± 0.003), and the lowest was found for TBRV-M1 (8.591 ± 0.003). In the case of hosts, the highest virus titre was observed for quinoa (estimated marginal mean ± 1 SD: 8.895 ± 0.003), then for tobacco (8.619 ± 0.003) and the lowest for spinach (8.499 ± 0.003) (Figure 3). The interaction between these two factors was also highly significant and of a large magnitude ($\eta_{P}^{2}$ = 0.988, *p* < 0.0001), meaning that differences in viral load among isolates was not always the same but depended on the host species in which the virus replicated. For example, TBRV-K8 accumulated the most in tobacco (8.817 ± 0.005) while TBRV-M1 and TBRV-P1 did so in quinoa (9.208 ± 0.005 and 8.770 ± 0.005, respectively) (Figure 3).

Moreover, the magnitude of the effect of satRNAs presence on TBRV viral load depends both on the viral isolate ($\eta_{P}^{2}$ = 0.866, *p* < 0.0001), host species ($\eta_{P}^{2}$ = 0.882, *p* < 0.0001), and the interaction of these two factors ($\eta_{P}^{2}$ = 0.936, *p* < 0.0001), all in a highly significant manner. While the presence of satRNAs reduced TBRV-K8 and TBRV-M1 loads (−19.5% and −16.6%, respectively), the effect was the opposite for TBRV-P1, for which the viral load increased (43.9%). The effect of satRNAs on viral load also differed between host plant species. While, on average, in quinoa and spinach, the presence of satRNAs increased the accumulation of TBRV (19.1% and 22.7%, respectively), in tobacco, the effect was the opposite, showing a −33.9% reduction (Figure 3).

Finally, the magnitude of this effect varied between isolate and host plant. For example, TBRV-K8 in the presence of satRNAs accumulated 21.9% more than in its absence in quinoa, and 46.5% less in spinach. TBRV-M1 accumulated 39.6% more in the presence of the satRNAs in spinach, but −67.8% less in tobacco. For TBRV-P1, the effect of satRNAs was always positive, but varied in magnitude from 7.4% in quinoa to 147.2% in spinach.

Regarding the progression of virus accumulation along infection, dpi also had a significant net effect (intra-subject effects: $\eta_{P}^{2}$ = 0.753, *p* < 0.0001) as well as in the interaction with all the above inter-subject factors and their interactions (in all cases $\eta_{P}^{2}$ ≥ 0.834, *p* < 0.0001). Regardless of the combination of factors, viral load showed a negative trend with dpi: marginal mean log-viral load at 7 dpi was 8.774 ± 0.003, while it declined to 8.587 ± 0.003 at 28 dpi (i.e., an overall −35.0% reduction).

### 2.5. Analysis of the Absolute Quantification of satRNAs

In this section, we will turn our attention to the statistical analysis of the satRNA accumulation. Absolute quantification data were fitted to the repeated measures that ANOVA described in Section 4.3.3 (Equation (2)). Following the same logic in the previous section, we first evaluated the effect of viral strain and host species in satRNA accumulation. SatRNA accumulation significantly varied among the three viral isolates ($\eta_{P}^{2}$ = 0.938, *p* < 0.0001), the magnitude of the effect being very large. SatRNA accumulation was larger in the presence of TBRV-P1 (estimated marginal mean of log-viral load: 11.187 ± 0.004), and lower when TBRV-M1 (11.003 ± 0.004) and TBRV-K8 (11.005 ± 0.004) acted as HVs (Figure 4). Focusing on host species, highly significant and large-in-magnitude differences were also observed ($\eta_{P}^{2}$ = 0.980, *p* < 0.0001), with quinoa accumulating more satRNA (11.283 ± 0.004) than spinach (10.985 ± 0.004) and tobacco (10.928 ± 0.004) (Figure 4). Interestingly, a highly significant interaction of large magnitude was also observed between these two inter-subject factors ($\eta_{P}^{2}$ = 0.975, *p* < 0.0001), indicating that the actual magnitude of the effect depends on the particular combination of host plant and HV isolate (Figure 4). For example, the maximum satRNA accumulation in quinoa was observed when the HV was TBRV-M1 (mean value 11.624 ± 0.007). However, the maximum satRNA accumulation in the other two host species was observed in the presence of TBRV-P1 (mean values 11.162 ± 0.007 in tobacco and 11.235 ± 0.007 in spinach, respectively). Likewise, the minimum satRNA accumulations in quinoa and tobacco were observed in the presence of TBRV-K8 (11.360 ± 0.007 and 11.078 ± 0.007, respectively), and in the presence of TBRV-M1 in spinach (11.167 ± 0.007).

Turning now to the intra-subject effects (i.e., differences in satRNA accumulation over time), highly significant overall differences were observed for individual plants along the duration of the experiment ($\eta_{P}^{2}$ = 0.969, *p* < 0.0001). However, in contrast with the monotonous decrease of accumulation observed for the genomic RNA, in the case of the satRNA, the temporal dynamic shows a minimum value of 7 dpi (10.805 ± 0.004), a maximum accumulation peak at 14 dpi (11.256 ± 0.004), and then a decrease down to 11.036 ± 0.004 at 28 dpi (Figure 5). The effect of time was also dependent on the interaction with the two inter-host factors discussed in the previous paragraph (in all cases $\eta_{P}^{2}$ ≥ 0.939, *p* < 0.0001). A detailed analysis of the interaction between dpi and host species and HV isolate shows that the overall trend of a peak in satRNA accumulation at 14 dpi was mostly due to the dynamics in spinach (log-viral load at 14 dpi was 11.278 ± 0.004), since tobacco shows a more erratic pattern and quinoa shows a smooth and constant decline from 7 dpi (11.575 ± 0.006) to 28 dpi (10.805 ± 0.006).

### 2.6. Association between Viral RNA and satRNA Accumulations

Next, we sought to evaluate the association between HV and satRNA accumulations. Figure 6 shows the observed relationships for each viral isolate and the three host species. Overall, a significant positive partial correlation, controlling for viral isolate and plant host species, exists (*r_p_* = 0.598, 175 d.f., *p* < 0.0001), suggesting that more viral replication results in more satRNA replication, as well. However, as Figure 6 illustrates, this positive association holds true for TBRV-K8 and TBRV-P1 isolates regardless of the host species (in all six cases *r* ≥ 0.701, 18 d.f., *p* < 0.001), but not for TBRV-M1, which shows no significant association between the accumulation of these two RNA species in either of the hosts tested (*r* ≤ 0.272, 18 d.f., *p* ≥ 0.2606), suggesting that TBRV-M1 interacts with satRNA in a different manner than the other two isolates.

## 3. Discussion

Satellite RNAs are supernumerary RNAs that can affect the accumulation and pathogenicity of their associated (helper) virus, at the same time, being dependent on the HV for their replication and infection [25]. Three types of the satRNAs have been distinguished so far: large, linear satRNAs (from 800 to 1500 nt), short, linear satRNAs (<700 nt), and short, circular satRNAs (220 to 388 nt), of which only the large, linear satellite RNAs encode their own protein, which can be used for the replication of the satRNAs [6,17,26].

In this paper, we conducted a global analysis of the population of large, linear satRNAs associated with different isolates of TBRV. We identified the presence of satRNAs in 18 out of 42 analysed samples in the Polish population of the virus. We have shown that obtained satellite RNAs of TBRV are diverse, with a sequence identity ranging from 91.1% to 100% in the case of nucleotides, and from 89.4% to 100% for amino acids. The greatest differences were observed for satRNAs of TBRV-C [22] and TBRV-O1 (this study), for which the nucleotide and amino acids sequences identities were 91.1% and 89.4%, respectively. In addition, we observed that some satRNAs differ in length due to the presence of an insertion or deletion within the 3′ UTR region. It has been previously shown that the 3′ UTR region is responsible for satRNAs virulence, and insertions or deletions close to the 3′ region can lead to a loss of biological activity [27]. In contrast, lack of genetic variation was observed in the 5′ UTR region of analysed sTBRV sequences. Additionally, for other closely related nepoviruses such as GFLV and ArMV, no genetic variation was found in the 5′ UTR region [19]. The origin of the satellite molecule has not been clearly defined so far, but it is speculated that it may be derived from viruses, viroids, or as a result of recombination between the HV and host mRNAs [28]. Phylogenetic and evolutionary inference can be often misinformed if recombination is not accounted for. When intragenic recombination occurs, different parts of the sequence have different phylogenetic histories; therefore, analyses of recombination events are an important stage of nearly every sequence-comparative study [24,29]. None of the statistical tests applied to our data set supported the existence of recombination events among satRNAs. This conclusion supports previous conclusions by Zarzyńska–Nowak et al. [15]. The fate of new genetic variants is largely determined by selection [30]. The evolution of satRNAs associated with TBRV isolates is driven both by positive and purifying selection, with the prevalence of the latter. Performed analyses showed that only 17 codons were under positive selection. As in our previous study, the selective pressure was mainly negative, indicating high genetic stability of the satRNAs population [15].

The overabundance of satRNAs associated with TBRV isolates originated from black locust in the population did not allow us to investigate the influence of the host plant in the placement of satRNAs in phylogenetic trees. Performed analyses showed that some satRNAs clustered together according to place of origin, however no unequivocal association between the genetic diversity of satellite particles and the collection region was confirmed. To investigate the degree of correlation between host/place of origin and satRNAs ancestry, a more heterogeneous (in case of this two traits), wider range of satRNAs sequences is needed. Performed phylogenetic analyses constitute extension of research conducted previously [15]. In our work, similar tendencies are presented: the greatest phylogenetic similarity was observed between isolates from tomato (TBRV-P1 and TBRV-Pi) and black locust (TBRV-M1 and TBRV-MJ). The satRNA molecule associated with the TBRV-C isolate, together with several satellite molecules obtained in this work, was outside the main cluster, forming a separate branch. This may be due to its belonging to a different serotype than the other satRNAs in the analysed population.

We also examined the symptom development and accumulation of gRNA of three isolates of TBRV with and without satRNA molecules in three different hosts, measured the accumulation of satRNA, and also performed a correlation analysis between the genomic and satellite RNAs. We observed that, regardless of the analysed isolate, for two hosts (quinoa and spinach), the symptoms were more severe in the presence of a satellite associated with genomic RNA. In both cases, we observed a slower growth rate and stronger changes in the leaf blades in the form of malformation, chlorosis, and necrosis. Only in the case of tobacco, the development of disease symptoms was similar regardless of whether the satellite particle was associated with genomic RNA or not. It has been previously found that in specific host plants, satRNAs can attenuate or exacerbate the symptoms induced by the HV. For example, infection of tomato with cucumber mosaic virus (CMV) associated with D-satRNA, B-satRNA, or WL1-satRNA induces necrosis, chlorosis, and attenuation, respectively [31]. The presence of systemic necrotic lesions on tomato after infection with CMV with satRNAs is associated with programmed cell death [31]. Necrogenic Y-satRNAs of CMV, which cause severe systemic necrosis in tomatoes, do not induce similar symptoms in other plants [32,33]. Moreover, Y-satRNAs induce necrosis in tomato when associated with the Y strain of CMV, but not with CMV strain O, and the symptoms also depend on the tomato cultivar [34].

We identified that the effect of satRNAs on the viral load is very complex and depends on different factors, including viral isolates, host species, and the interaction between them. While in the case of some isolates, the presence of satRNAs can reduce the viral load (TBRV-K8, TBRV-M1), for some others, it can increase the viral load (TBRV-P1). The isolates tested in this study came from various host plants, including economically important plants such as tomato and zucchini, but also from trees. We observed that satRNAs of TBRV-M1 originated from black locust decreased the accumulation of genomic RNA the most. We presumed that the different origins of the individual isolates tested in this experiment were related to different virus accumulation with and without the satellite particle. Interestingly, in the case of the hosts, the effect of the satellite on the virus load also differed between host plant species. While the accumulation of gRNA of TBRV with satRNAs was rather high in quinoa and spinach, in tobacco, the effect was the opposite, which was consistent with the symptoms in infected plants. It has been previously found that the effect of satRNAs on CMV replication, pathogenesis, and symptom development depends on the host plant and CMV strain [32]. The effect of satellite particles on their host might be related to interactions with the host plant factors, leading to changes in the genetic program involved in basic metabolism, plant defence, and some pathways important for plant development, and consequently, with changes in viral accumulation and disease symptoms in infected plants [5,35,36,37].

## 4. Materials and Methods

### 4.1. Analysis of the Genetic Diversity of satRNAs Population

#### 4.1.1. Virus Isolates

For the experiments, 42 isolates of TBRV collected between 1998 and 2020 in different regions of Poland were used. The isolates came from different hosts, both cultivated and ornamental plants, trees, and shrubs. All isolates were transferred by mechanical inoculation in *C. quinoa* and *N. tabacum* cv. Xanthi and maintained under greenhouse conditions (22–23 °C, 16/8 h photoperiod). After seven days post inoculation (dpi), the apical leaves of the infected plants were harvested and used for RNA isolation. Viral RNAs were extracted by a phenol-chloroform procedure, measured spectrophotometrically using ND-2000 spectrophotometer (Thermo Fisher Scientific, Wilmington, DE, USA), and diluted to the final concentration of 1 μg/μL.

#### 4.1.2. Amplification, Cloning and Sequencing of Satellite RNAs

First strand cDNAs were synthesized using 1 μL of each RNAs at the final concentration of 1 μg/μL, sat_insert_R [15] and SuperScript™ II Reverse Transcriptase (Thermo Fisher Scientific, Waltham, MA, USA) according to the manufacturer’s protocol. The satRNAs were amplified using two sets of reagents: Expand™ Long Range dNTPack (Roche, Basel, Switzerland) and CloneAmp™ HiFi PCR Premix (Takara Bio Inc., Shiga, Japan) and primers pair: sat_insert_F and sat_insert_R designated by Zarzyńska-Nowak et al. [15]. PCRs were carried out in a Thermal Cycler (Biometra, Goettingen, Germany) and consisted of: an initial denaturation at 92 °C for 2 min, followed by 25 cycles of 10 s at 92 °C, 15 s at 65 °C, and 60 s/ kb at 68 °C followed by a 10 min at 72 °C (for the first set) or 35 cycles of 10 s at 98 °C, 15 s at 55 °C, and 5 s/kb at 72 °C (for the second set). The resulting PCR products were separated on a 1 % agarose gel to verify the appropriate size of the amplicons. All products were cloned into pCR^®^ 4-TOPO^®^ Vector (Thermo Fisher Scientific, Waltham, MA, USA), or CloneJET PCR Cloning Kit (Thermo Fisher Scientific, Waltham, MA, USA) and plasmids were transformed into *Escherichia coli,* according to manufacturer’s instructions. Plasmid DNAs were isolated from individual colonies using the GeneJET Plasmid Miniprep Kit (Thermo Fisher Scientific, Waltham, MA, USA) according to the manufacturer’s protocol. The presence of the insert was checked by restriction cleavage using BglII, NotI, and EcoRI enzymes (Thermo Fisher Scientific, Waltham, MA, USA) and sequenced by external company using the standard Sanger procedure (Genomed S.A., Warsaw, Poland).

#### 4.1.3. Recombination and Selective Pressure Analysis

The obtained full-length sequences of 18 TBRV satRNAs were edited and compiled in the BioEdit Sequence Alignment Editor [38], then compared with 8 other satRNA sequences deposited in the GenBank. For further analyses, coding region of sequences were aligned by codons using MUSCLE, as implemented in MEGA X [21].

Prior to that, the analysed population of satRNAs was tested for the presence of recombination signals with RDP4 v. 4.100 software: GENCONV, BootScan, MaxChi, Chimaera, SiScan, 3Seq and RDP [23]. The occurrence of a recombination was considered statistically significant if five or more methods had a *p* < 0.05. Additionally, to investigate the presence of recombination breakpoints in our data, the Datamonkey Adaptive Evolution Server with the Genetic Algorithm for Recombination Detection (GARD) method was used [24].

Subsequently, the selective pressure affecting individual codons in the satRNA coding sequences was assessed based on the ratio of non-synonymous to synonymous substitutions (*ω = d_N_*/*d_S_*). The analysis was performed using the Datamonkey Adaptive Evolution Server using the following algorithms: Fixed Effects Likelihood (FEL), Fast Unconstrained Bayesian Approximation (FUBAR), Single Likelihood Ancestor Counting (SLAC), and Mixed Effects Model of Evolution (MEME) [39]. The significance value was set to *p* < 0.05 for the FEL, SLAC, and MEME. For the FUBAR method, the significance level was determined for the BPP > 0.9.

#### 4.1.4. Phylogenetic Analysis of satRNAs

Phylogenetic analysis was performed using MEGA X software, based on the coding sequence of 26 TBRV satRNAs, both obtained in this experiment, as well as those deposited in GenBank. The information about particular isolates (name of the satRNAs, host plant and country/region of origin) were placed on the tree. The phylogenetic tree was constructed using the maximum likelihood algorithm (ML) and the two-parameter Kimura model with gamma distribution (K2 + G), which has been assessed as the best model of nucleotide substitution in MEGA X. In addition, based on the amino acid sequences, a second tree was constructed using the ML algorithm and Jones–Thornton–Taylor model (JTT). For both trees, bootstrap values were calculated using 1000 random pseudoreplicates. The obtained phylogenetic trees were edited and visualized in the Evolview online platform [40].

### 4.2. Obtaining an Infectious Clone of satRNA

An infectious clone of the satRNA molecule was obtained for three TBRV isolates: TBRV-M1, TBRV-K8 and TBRV-P1. Since our previous study indicated that nucleotide sequences of satRNA of these isolates are identical [15], only one (satRNAs of TBRV-P1 isolate) was used in this study. The full-length sequence of TBRV-P1 satRNA was amplified with the primers: sat_insert_F and sat_insert_R [15], using CloneAmp HiFi PCR Premix (Takara Bio Inc., Kusatsu, Japan), as described above. The obtained product was flanked at the 5′ and 3′ ends by 20 nucleotide sequences, providing the homology to 5′ and 3′ ends of linearized binary vector pJL89 [41]. A backbone cloning vector was obtained using CloneAmp HiFi PCR Premix and pJL89.F/pJL89.R primes [42]. In the next step, purified PCR products were cloned into pJL89 between double cauliflower mosaic virus (CaMV) 35S promoter and the hepatitis delta virus (HDV) ribozyme followed by the NOS terminator. In-Fusion reaction and transformation into Stellar Competent cells (Takara Bio Inc., Kusatsu, Japan) were performed using the In-Fusion HD Cloning Kit (Takara Bio Inc. Kusatsu, Japan) according to the manufacturer’s protocol. The obtained plasmid DNAs were purified using GeneJET Plasmid Miniprep Kit (Thermo Fisher Scientific, Waltham, MA, USA) and sequenced to confirm the presence of the insert (Genomed S.A., Warsaw, Poland). Then, the plasmids were transferred into *Agrobacterium tumefaciens* (strain GV3101), according to a previously used procedure [42]. Simultaneously, previously obtained full-length cDNA infectious clones of RNA1 and RNA2 of TBRV-M1, TBRV-P1, and TBRV-K8 isolates were used [15,42]. *Nicotiana benthamiana* Domin. The plants were agroinfiltrated using RNA1 and RNA2 of each isolate and satRNAs were obtained at a ratio of 1:1:1 and maintained under greenhouse conditions (23°C, 16/8 photoperiod). The presence of genomic RNA (gRNA) and satRNAs in agroinfiltrated plants was checked after 10 days post agroinfiltration (dpa) by standard RT-PCR procedure with the primers sat_insert_F/R and TBRV CPF/R [43].

### 4.3. Accumulation Experiments

#### 4.3.1. Absolute Quantification of gRNA by RT-qPCR

For the quantification experiments, three isolates originated from different host plants: TBRV-K8, TBRV-M1, and TBRV-P1, were selected. *N. benthamiana* plants were agroinfiltrated using TBRV infectious copies with or without satRNAs, respectively, and maintained under greenhouse conditions (23 °C, 16/8 photoperiod). 14 dpa 500 mg of apical leaves were harvested and crushed in 2 mL of phosphate buffer (0.05 M, pH 7.2). The obtained inoculum of each isolate with or without satRNAs was used to inoculate the following test plants: quinoa, tobacco, and spinach. All plants were grown under greenhouse conditions in closed, monitored compartments for 28 dpi. In parallel, plants inoculated only with phosphate buffer were grown as a negative control.

The RNA accumulation level and symptoms development were analysed for each studied variant separately in five biological replicates. After 7, 14, 21, and 28 dpi, viral RNAs were isolated from each individual plant separately using a phenol-chloroform procedure, measured fluorometrically using Qubit 3 (Thermo Fisher Scientific, Waltham, ME, USA), and diluted to the final concentration of ca. 10 ng/μL. To prepare a standard curve, RNA transcripts of the CP gene of each isolate (TBRV-M1, TBRV-K8, and TBRV-P1) were obtained according to the previously described procedure [4]. RNA transcripts were produced using the mMESSAGE mMACHINE Kit (Thermo Fisher Scientific, Waltham, ME, USA) according to the manufacturer’s protocol, measured and diluted to the final concentration of 1 μg/μL, which corresponded to 1.183 × 10^12^ copies of viral genomes. Then, the obtained RNA transcripts were 10-fold serial diluted from 1.183 × 10^12^ to 1.183 × 10^7^ copies of viral genomes, using appropriate RNAs from healthy plants as diluents. RT-qPCR reaction was performed using iTaq SYBR Green (Biorad, Hercules, CA, USA) and CPqF/R primers [4]. Each sample was analysed in three technical replicates. The number of viral genomes in each sample after 7, 14, 21, and 28 dpi was calculated by comparing the obtained results to the values from the standard curve with LightCycler^®^ 96 SW 1.1 software (Roche, Mannheim, Germany).

#### 4.3.2. Absolute Quantification of satRNAs by RT-qPCR

In all samples previously inoculated with isolates: TBRV-K8, TBRV-M1 and TBRV-P1 associated with satRNAs, the accumulation of satRNA molecules was determined. To prepare a standard curve, RNA transcripts of satRNA were obtained. First, the satRNA was amplified using plasmid clone with the primers: TBRVSATpromotor: 5′ ATGAATTCTAATACGACTCACTATAGTTTGAAAATAATTTTGAAAGTCTCTGACAATCGTAACTGATG 3′ and TBRVsatTRANSKRYPTR: 5′ GCGGCCGCTATTAATTTGCTCTGGAGAAAAGGTATACAATCT (this study), at the annealing temperature of 54 °C. The obtained PCR product was then cut using NotI restriction enzyme, verified by separating on 1% agarose gel and then purified using NucleoSpin^®^ Gel and PCR Clean-up (MACHEREY-NAGEL, Dueren, Germany), according to the manufacturer’s protocol. RNA transcript was produced using the mMESSAGE mMACHINE Kit (Thermo Fisher Scientific, Waltham, MA, USA) according to the manufacturer’s protocol, measured and diluted to the final concentration of 1 μg/μL, which corresponded to 1.328 × 10^12^ copies of viral genomes. The obtained transcript of satRNA was then 10-fold diluted from 1.328 × 10^12^ to 1.328 × 10^7^ copies of viral genomes. RT-qPCR reaction was performed using iTaq SYBR Green (Biorad, Hercules CA, USA), and the primers: RACE3sat 5′ GCTGTCCCTTGCTTACTTCA 3′ [15] and SATrealNewR2 5′ CACGCTGCACCTTCATA 3′ (this study). The reaction was performed at the annealing temperature of 54 °C in three technical replicates each, according to the manufacturer’s protocol. The number of viral genomes in each sample after 7, 14, 21, and 28 dpi was calculated by comparing the obtained results to the values from the standard curve with LightCycler^®^ 96 SW 1.1 software (Roche, Mannheim, Germany).

#### 4.3.3. Statistical Analysis

TBRV gRNA quantifications (*Q*) were fitted to a Repeated Measures ANOVA in which the HV isolate (*V*), plant host (*H*), and presence/absence of satRNA (*S*) were considered as inter-subject orthogonal fixed factors, and independent plants (*P*) as a random factor nested within the interaction of the three other factors (*V* × *H* × *S*). Each plant was repeatedly observed at four different time points (*t*), with each measure being considered as the intra-subject factor. The model equation reads as:(1)$$Q_{ijklt}\sim q+t+V_{it}+H_{jt}+S_{kt}+{\left(V\times H\right)}_{ijt}+{\left(V\times S\right)}_{ikt}+{\left(H\times S\right)}_{jkt}+{\left(V\times H\times S\right)}_{ijkt}+P{\left(V\times H\times S\right)}_{ijklt}+\varepsilon_{ijklt}$$
where, *q* represents the grand mean gRNA quantification, and *ε* the error term, assumed to be Gaussian.

SatRNA quantifications (*R*) were fitted to a second Repeated Measures ANOVA with the following model equation:(2)$$R_{ijkt}\sim r+t+V_{it}+H_{jt}+{\left(V\times H\right)}_{ijt}+P{\left(V\times H\right)}_{ijkt}+\varepsilon_{ijkt}$$
where, *r* represents the grand mean satRNA quantification and the other terms are as in Equation (1).

For these two models, the size of the effects was evaluated using the $\eta_{P}^{2}$ statistic that measures the proportion of variance explained by a given factor of the total variance remaining after accounting for variance explained by other factors in the model. Usually, $\eta_{P}^{2}$ > 0.15 indicate large effects.

All statistical analyses were performed with SPSS version 27 software (IBM, Armonk, NY, USA).

## 5. Conclusions

In conclusion, we identified a population of 18 satRNAs in the TBRV population circulating in Poland, which was characterized by a large genetic diversity. A strong effect of selective pressure was also demonstrated, which proves the removal of unfavourable mutations from the population. Our data provide evidence that the presence of satRNAs associated with a viral genome significantly impact TBRV accumulation in an isolate and host-dependent manner. Moreover, a positive correlation between the accumulation of gRNA and satRNAs for two of the analysed isolates was demonstrated.

## Figures and Tables

**Figure 1 ijms-23-09393-f001:**
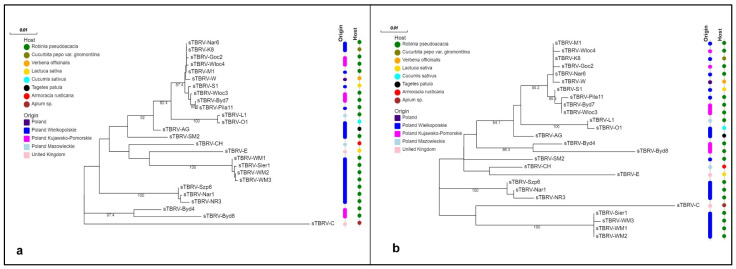
Maximum likelihood phylogenetic trees constructed based on (**a**) nucleotide and (**b**) amino acid sequences of 26 satRNAs associated with TBRV. Trees were generated with MEGA X. See Section 4.1.4 for details.

**Figure 2 ijms-23-09393-f002:**
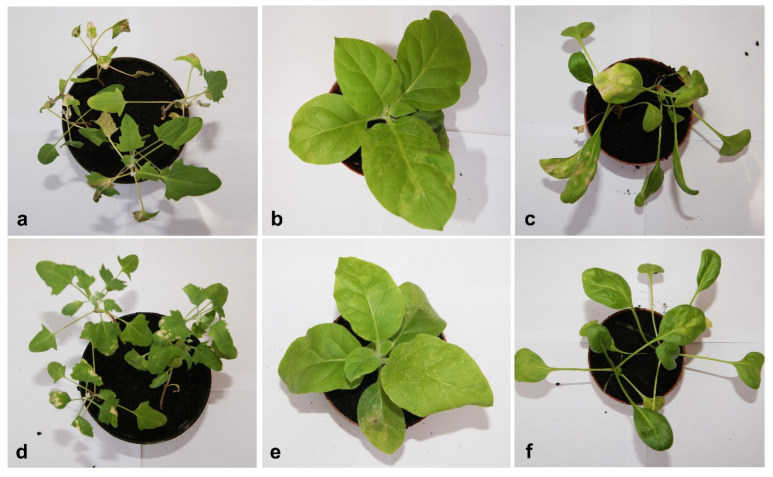
Disease symptoms observed on host plants after 10 dpi: quinoa (**a**,**d**), tobacco (**b**,**e**) and spinach (**c**,**f**) infected with TBRV-K8 isolate with (**a**–**c**) and without (**d**–**f**) satRNAs.

**Figure 3 ijms-23-09393-f003:**
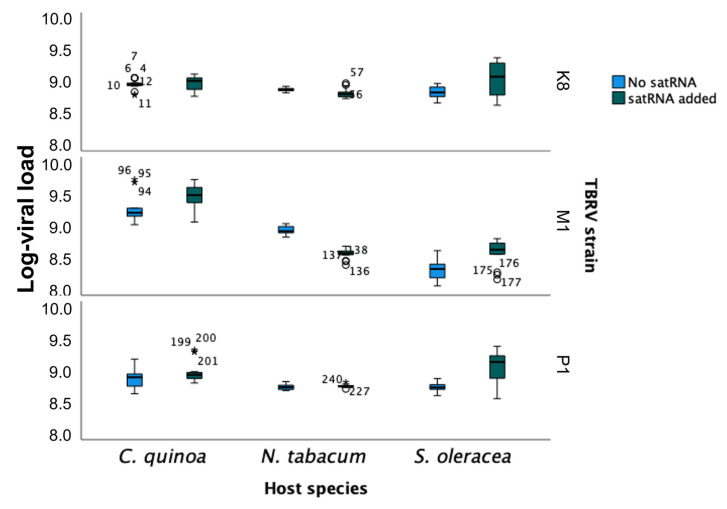
The effect of satRNAs presence on the accumulation of three TBRV isolates (TBRV-K8, TBRV-M1, TBRV-P1) in different host plants: quinoa, tobacco, and spinach. Boxes represent interquartile ranges (IQR). Horizontal lines within the boxes represent the median value and error bars the 95% confidence interval of the mean. Dots represent atypical cases (above or below 1.5 × IQR) and stars represent extreme cases (above or below 3 × IQR).

**Figure 4 ijms-23-09393-f004:**
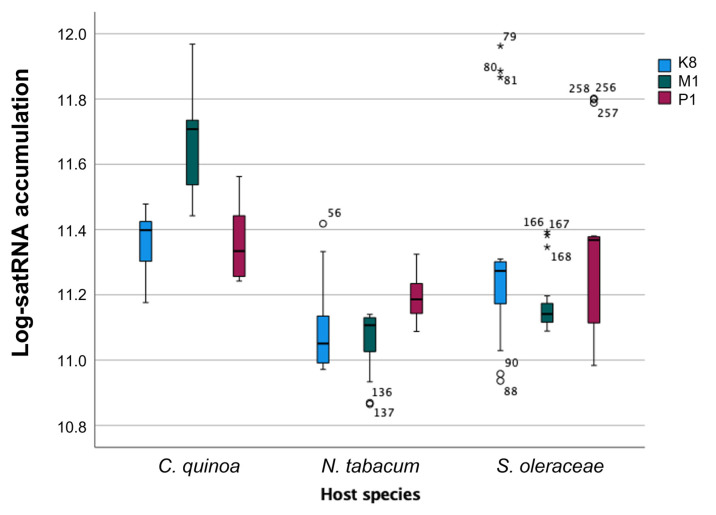
Accumulation of satRNAs in presence of three different isolates of TBRV (TBRV-K8, TBRV-M1, TBRV-P1) acting as HV and in three different host plants: quinoa, tobacco, and spinach. Boxes represent interquartile ranges (IQR). Horizontal lines within the boxes represent the median value and error bars the 95% confidence interval of the mean. Dots represent atypical cases (above or below 1.5 × IQR) and stars represent extreme cases (above or below 3 × IQR).

**Figure 5 ijms-23-09393-f005:**
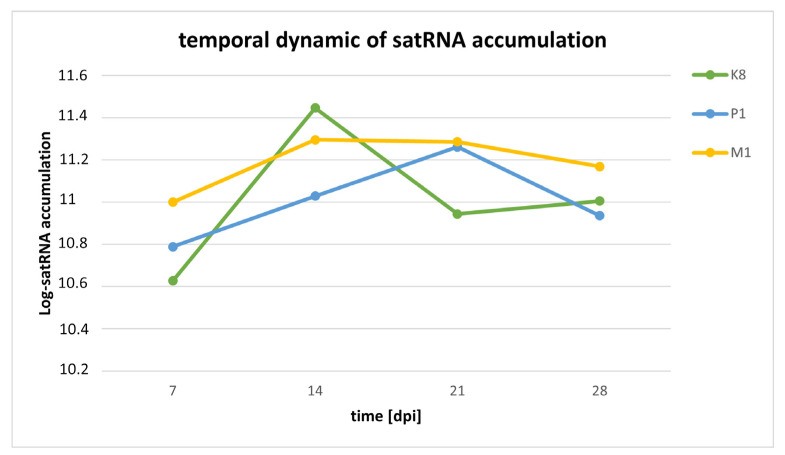
Dynamic of satRNA accumulation over time.

**Figure 6 ijms-23-09393-f006:**
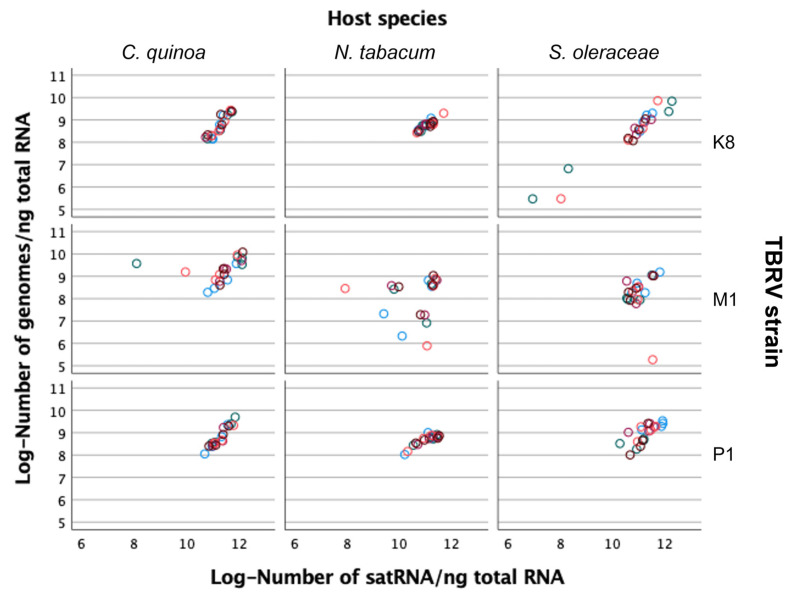
Relationship between viral RNA genome and satRNA accumulations for each HV isolate and host species. Different colored dots represent the four time points taken for each of the independent plant replicates (*n* = 5).

**Table 1 ijms-23-09393-t001:** satRNAs of 18 TBRV isolates obtained in this study, and of eight additional TBRV isolates retrived from GenBank. The individual columns contain: the name of the satRNAs, host plant, location, GenBank accession numbers and reference.

satRNA	Host	Location	Accession	Reference
sTBRV-Byd4	*Robinia pseudoacacia*	Wielkopolska, Poland	ON953201	this study
sTBRV-Byd7	*R. pseudoacacia*	Wielkopolska, Poland	ON953202	this study
sTBRV-Byd8	*R. pseudoacacia*	Wielkopolska, Poland	ON953203	this study
sTBRV-CH	*Armoracia rusticana*	Mazowieckie, Poland	ON953204	this study
sTBRV-Goc2	*R. pseudoacacia*	Kujawsko-pomorskie, Poland	ON953205	this study
sTBRV-L1	*R. pseudoacacia*	Mazowieckie, Poland	ON953206	this study
sTBRV-Nar1	*R. pseudoacacia*	Wielkopolska, Poland	ON953207	this study
sTBRV-Nar6	*R. pseudoacacia*	Wielkopolska, Poland	ON953208	this study
sTBRV-NR3	*R. pseudoacacia*	Wielkopolska, Poland	ON953209	this study
sTBRV-O1	*Cucumis sativus*	Wielkopolska, Poland	ON953210	this study
sTBRV-P11	*R. pseudoacacia*	Wielkopolska, Poland	ON953211	this study
sTBRV-Sier1	*R. pseudoacacia*	Wielkopolska, Poland	ON953212	this study
sTBRV-Szp6	*R. pseudoacacia*	Wielkopolska, Poland	ON953213	this study
sTBRV-Wloc3	*R. pseudoacacia*	Kujawsko-pomorskie, Poland	ON953214	this study
sTBRV-Wloc4	*R. pseudoacacia*	Kujawsko-pomorskie, Poland	ON953215	this study
sTBRV-WM1	*R. pseudoacacia*	Wielkopolska, Poland	ON953216	this study
sTBRV-WM2	*R. pseudoacacia*	Wielkopolska, Poland	ON953217	this study
sTBRV-WM3	*R. pseudoacacia*	Wielkopolska, Poland	ON953218	this study
sTBRV-K8	*Cucurbita pepo*	Wielkopolskie, Poland	MN699709	[15]
sTBRV-M1	*R. pseudoacacia*	Wielkopolska, Poland	MN699710	[15]
sTBRV-W	*Verbena officinalis*	Mazowieckie, Poland	MN718462	[15]
sTBRV-S1	*Lactuca sativa*	Wielkopolska, Poland	MN699711	[15]
sTBRV-AG	*Tagetes patula*	Wielkopolska, Poland	MN699708	[15]
sTBRV-SM2	*R. pseudoacacia*	Wielkopolska, Poland	MN699711	[15]
sTBRV-E	*L. sativa*	Norfolk, UK	X05688.1	[22]
sTBRV-C	*Apium* sp.	Suffolk, UK	X056 89.1	[22]

**Table 2 ijms-23-09393-t002:** Summary of the results of the selective pressure analysis obtained by using the algorithms implemented in the Datamonkey server. The codons detected by the three methods are marked in green. Blue shows common codons for FUBAR and MEME, red shows common codons for FEL and FUBAR.

FUBAR		FEL		MEME	SLAC		Codon Position
*ω* > 1	*ω* < 1	*ω* > 1	*ω* < 1	*ω* > 1	*ω* > 1	*ω* < 1	19, 52, 74, 99, 114, 125, 162, 169, 174, 208, 224, 227, 244, 245, 300, 313, 351
16	44	5	41	7	-	14	

## Data Availability

All sequences of satellite RNAs have been deposited in the GenBank under accession numbers ON953201–ON953218.

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
