# Peer review of "Genetic Diversity of Tomato Black Ring Virus Satellite RNAs and Their Impact on Virus Replication"

_ijms, 2022, doi:10.3390/ijms23169393_

Round 1

Reviewer 1 Report

In this manuscript, the authors characterize viral satRNAs associated with tomato black ring virus (TBRV).  The identified several variants in existing populations and that deletion of these satRNAs could impact viral replication and disease across three host species.  Overall, the paper is interesting and the experiments are well-performed.  There are however a few things that would be helpful in asserting the findings:

As the satRNA sequencing involves RT and PCR amplification, Taqman qPCR validation of the single point-mutations would be helpful to rule out errors introduced during cloning.

The largest accumulation is also observed in quinoa in this figure, while the number stated in the text is below the cutoff of the axis, could this not be 11.674 and plotted for tobacco?

The paragraph from 208 to 221 has interesting data and should be made into a figure for visualization

Can the defects in growth over time or other quantifiable disease symptoms be plotted in addition to the images shown?  This would be helpful in establishing their impact.

There are also a few minor text errors:

Lines 124, 130, 134, 275-276, 284-285, 313 change locust to locus

line 145 comma after quinoa

line 147: In tobacco (comma) chlorotic ringspots and mosaic were visible (Figure 2b, e),(change comma to semicolon) and in spinach (comma) chlo- Figure 3: log scale makes it difficult to see differences, could this be more obvious on linear graphs? line 191 comma after section line 193 remove than line 196: the magnitude of the effect being very large line 207: while for TBRV-M1 the maximum accumulation happened in tobacco (10.674... line 268: suggested by (the) constructed phylogenetic network,(remove comma) were not statistically significant line 283: comma after work line 294: comma after cases Line 310: comma after -M1) Line 313: from the black locus

Author Response

We are very thankful for all your comments and suggestions. We revised manuscript taking into account your specific comments listed. We prepare additional figure representing the dynamic of satRNA accumulation along time. 

Comments:

In this manuscript, the authors characterize viral satRNAs associated with tomato black ring virus (TBRV).  The identified several variants in existing populations and that deletion of these satRNAs could impact viral replication and disease across three host species.  Overall, the paper is interesting and the experiments are well-performed.  There are however a few things that would be helpful in asserting the findings:

As the satRNA sequencing involves RT and PCR amplification, Taqman qPCR validation of the single point-mutations would be helpful to rule out errors introduced during cloning.

Authors: Thank you for your suggestion. However, we believe that the amplification and cloning procedure used in these experiments allowed to obtain reliable results and there is no need to use additional checking methods. For each isolate we sequenced 3 clones. The obtained sequences were collected and compared using Bioedit software and the consensus sequences were selected.

The largest accumulation is also observed in quinoa in this figure, while the number stated in the text is below the cutoff of the axis, could this not be 11.674 and plotted for tobacco?

Authors: We revised this part of the manuscript.

The paragraph from 208 to 221 has interesting data and should be made into a figure for visualization

Authors: We added the figure according to your suggestion.

Can the defects in growth over time or other quantifiable disease symptoms be plotted in addition to the images shown?  This would be helpful in establishing their impact.

Authors: In our experiment, we did not measure the growth of individual plants after 7, 14, 21 and 28. We only conducted observations to assess the differences between satRNA-containing and non-satRNA variants. Therefore, preparation of plots is not possible. At the same time, we believe that the differences visible in the photos show the real impact of the presence of the satellite on the occurrence of disease symptoms.

There are also a few minor text errors:

Lines 124, 130, 134, 275-276, 284-285, 313 change locust to locus

Authors: The English name of the R. pseudoacacia is black locust.

line 145 comma after quinoa

Authors: We added it.

line 147: In tobacco (comma) chlorotic ringspots and mosaic were visible (Figure 2b, e),(change comma to semicolon) and in spinach (comma) chlo- 

Authors: We changed it.

Figure 3: log scale makes it difficult to see differences, could this be more obvious on linear graphs? 

Authors: In preparing Figures 3 and 4, we did not change the logarithmic scale to a linear one because in our opinion the logarithmic scale sufficiently emphasize the difference between the isolates and hosts.

line 191 comma after section line 193 remove than line 196: the magnitude of the effect being very large line 207: while for TBRV-M1 the maximum accumulation happened in tobacco (10.674... line 268: suggested by (the) constructed phylogenetic network,(remove comma) were not statistically significant line 283: comma after work line 294: comma after cases Line 310: comma after -M1) Line 313: from the black locus

 Authors: This sentence has been entirely reformulated.

Reviewer 2 Report

The manuscript is well written and exposed. I do not have great points to be raised, just few comments that are included in the attached file. 

Author Response

We are very thankful for all your comments and suggestions. We revised manuscript taking into account your specific comments listed. 

Comments:

Line 64 add reference

 Authors: We added the reference.

Line 67: a curiosity: there is a correspondence with the presence of consensus in 5’UTR and nepovirus classification in group A, B and C?

Authors: To our knowledge, the separation for A, B and C group was made based on the RNA2 size, phylogenetic relationships of the coat protein gene and cleavage site specificity of the protease.

Line 67: line 68-72 should be moved above when it is reported that are no data about the role of sat RNA in pathogenesis or move line 55 here

Authors: We moved line 55 into this paragraph.

Line 141: as the experimental trial is based on different time-points (line 428: 7, 14, 21 and 28 dpi) why is not included an analysis of the effects of RNAsat in these time points. It there was no effect a sentence could be included to state it.

Authors: Our analysis of the effect of satRNA on viral load is based on data obtained from different time points (7, 14, 21 and 28 dpi). Detailed information on viral load at individual time points can be found in the last part of the paragraph (lines 142-158): 'Regardless the combination of factors, viral load showed a negative trend with dpi: marginal mean log-viral load at 7 dpi was 8.774 ± 0.003, while it went down to 8.587 ±0.003 at 28 dpi (i.e., overall –35.0% reduction). '

Line 279: it could be interesting to double check if the isolates where the satRNA were found have the same clustering pattern or not.

Authors: We compared the results from phylogenetic analysis of satRNAs and TBRV CP (BudzyÅ„ska et al. 2021, Plant Pathology DOI: 10.1111/ppa.13382). We did not observe the same clustering pattern for isolates where satRNAs were found.